# Synchronized spatial shifts of Hadley and Walker circulations

Kyung-Sook Yun[1,2*], Axel Timmermann[1,2], and Malte F. Stuecker[3]

[1]Center for Climate Physics, Institute for Basic Science (IBS), Busan 46241, South Korea
[2]Pusan National University, Busan 46241, South Korea
[3]Department of Oceanography and International Pacific Research Center, School of Ocean and Earth Science and Technology, University of Hawai'i at Mānoa, Honolulu, HI, USA

*Correspondence to*: Kyung-Sook Yun (kssh@pusan.ac.kr)

**Abstract.** The El Niño-Southern Oscillation (ENSO) influences the most extensive tropospheric circulation cells on our planet, known as Hadley and Walker circulations. Previous studies have largely focused on the effect of ENSO on the strength of these cells. However, what has remained uncertain is whether interannual sea surface temperature anomalies can also cause synchronized spatial shifts of these circulations. Here, by examining the spatio-temporal relationship between Hadley and Walker cells in observations and climate model experiments, we demonstrate that the seasonally evolving warm pool sea surface temperature (SST) anomalies in the decay phase of an El Niño event generate a meridionally asymmetric Walker circulation response, which couples the zonal and meridional atmospheric overturning circulations. This process, which can be characterized as a phase-synchronized spatial shift in Walker and Hadley cells, is accompanied by cross-equatorial northwesterly low-level flow that diverges from an area of anomalous drying in the western North Pacific and converges towards a region with anomalous moistening in the southern central Pacific. Our results show that the SST-induced concurrent spatial shifts of the two circulations are climatically relevant as they can further amplify extratropical precipitation variability on interannual timescales.

## 1 Introduction

Changes in the zonal equatorial Walker cell (WC; tropical-mean zonal cell) and the meridional Hadley Cell (HC; zonal-mean meridional cell) are known to cause major climate disruptions across our planet. Because of their considerable impacts on various regional climates and extreme events, such as heat waves (Garcia-Herrera et al., 2010), tropical cyclones (Wu et al., 2018), sea level rise in the western Pacific (Timmermann et al., 2010), droughts (Dai, 2011;Lau and Kim, 2015), and regional monsoon variability (Kumar et al., 1999;Bollasina et al., 2011), the variations in the strength and position of WC and HC have been examined extensively across a wide range of timescales. It has been shown that changes in the strength of the WC are connected to those of the HC, in part due to the shared ascending branch of zonal and meridional overturning cells in the warm pool region (Vecchi and Soden, 2007;England et al., 2014;Liu and Zhou, 2017;Ma et al., 2018;Klein et al., 1999;Karnauskas and Ummenhofer, 2014). On interannual timescales, the co-variability between WC and HC strengths is tied to the El

Niño/Southern Oscillation (ENSO) related sea surface temperature (SST) gradients along the equator, associated with uneven spatial distribution of tropical convection (Oort and Yienger, 1996;Klein et al., 1999;Minobe, 2004). However, observational analyses suggest more complicated relationships, that cannot be fully explained by peak ENSO dynamics alone (Clarke and Lebedev, 1996;Mitas and Clement, 2005;Tanaka et al., 2005;Ma and Zhou, 2016).

35

In general, these two large-scale circulations can change independently from each, either in terms of their strength or their geographical position, but during strong El Niño and La Niña winters these circulations can change in unison. More specifically during peak El Niño events in boreal winter, the WC weakens and the rising branch shifts eastward. This is accompanied by a strengthening of the HC and an enhanced upward motion in the equatorial region (Ma and Li, 2008;Bayr et al., 2014;Guo and Tan, 2018;Minobe, 2004;Klein et al., 1999) (also see Supplementary text, Fig. S1, and Table S1). For peak La Niña conditions in winter the atmospheric response is approximately opposite. Previous studies have mostly focused on the response of the HC and WC strength to the peak phase of ENSO (i.e., an weakening of WC and strengthening of HC). Here we address a different question: Under what circumstances do the WC and HC show concurrent shifts in their geographic position? We further investigate whether these two dominant atmospheric circulation cells are coupled even in the absence of tropical ENSO-related SST anomalies. To address these questions, we conduct a comprehensive analysis of the dynamical coupling between HC and WC using observational data and a series of SST-forced atmospheric general circulation model (AGCM) simulations. We will focus on two important degrees of freedom that characterize variations of these circulations - their strength and spatial position.

## 2 Data and Method

### 2.1 Observations

We used the monthly reanalysis circulation dataset from European Centre for Medium-Range Weather Forecasts (ECMWF) atmospheric interim (ERAI) data from 1979 to 2017 (Dee et al., 2011). In addition, we obtained the monthly SST and precipitation data from the extended reconstruction of global SST (ERSSTv5) since 1854 (Smith et al., 2008) and Global Precipitation Climatology project (GPCP) version 2.3 from 1979 to 2017 (Adler et al., 2003), as well as the Climatic Research Unit (CRU) TS4.03 land precipitation from 1901 to 2017 (Harris et al., 2014), respectively. The monthly anomalies for 1979-2017 were calculated by removing the seasonal cycle and linear trend coefficient during the analysis period. The ENSO variability was characterized by a spatial average of SST anomalies over the tropical eastern Pacific Niño3 region [5°S-5°N, 150°W-90°W].

### 2.2 AMIP and CMIP models

We used 40 AGCM simulations from the forced AGCM Intercomparison Projection (AMIP): 19 model runs are part of the Coupled Model Inter-comparison Project (CMIP) Phase 5 (Taylor et al., 2012) and 21 models are part of Phase 6 (Eyring et al., 2016) (i.e., AMIP5 and AMIP6; Supplementary Tables S2 and S3). The AMIP simulations were forced by the observed SST and sea ice concentrations from 1979-2008 for AMIP5 and from 1979-2014 for AMIP6. The AMIP runs use the observed SST boundary forcing, which allow us to directly compare SST-forced atmospheric responses in simulations and observations. The AMIP multi-model ensemble (MME) represents the SST-forced signal, whereas a mixed signal of SST-forced variability and unforced atmospheric noise that is present in the observations and individual AMIP simulations. Additionally, to examine the impact of SST forcing amplitude on the WC-HC synchronization, we analyzed a set of 40 multi-model CMIP5 historical simulations covering the industrial period from 1900-2005 (see (Taylor et al., 2012) for details). Only one ensemble member (r1i1p1) was used for each model. All observations and MME data were interpolated to a regular 2.5° × 2.5° horizontal grid before the analyses.

## 2.3 Walker cell and Hadley cell variability

We calculated the monthly mass stream function (MSF) anomalies on the three-dimensional atmospheric circulation to describe the WC and HC circulations:

$$\psi_{WC}(x,z,t) = \frac{2\pi a}{g} \int_p^{p_s} u_D dp \quad , \tag{1}$$

$$\psi_{HC}(y,z,t) = \frac{2\pi a cos\phi}{g} \int_p^{p_s} [v] dp \quad , \tag{2}$$

Following previous work (Yu and Zwiers, 2010), the tropical zonal circulation ($\psi_{WC}$) is expressed as the divergent component ($u_D$) of zonal wind averaged over the tropics [5°S-5°N] (zonal mass-stream function). The tropical meridional circulation ($\psi_{HC}$) is the zonally averaged meridional mass-stream function. Next, we conducted an Empirical Orthogonal Function (EOF) analysis of monthly MSF anomalies to characterize the dominant modes of the tropical atmospheric mass flux variability. For fair comparison between observations and simulations, the MSF fields in every single model were projected onto the EOF pattern derived from the observations to generate PC time series for the simulated WC and HC variability. The inter-annual component of variability was isolated by applying a 10-point Butterworth band-pass filter with 1.5- year (18-months) and 10-year cutoffs. The WC and HC strengths were respectively calculated as the maximum value of tropical mean [5°S-5°N] MSF at 500 hPa and as the maximum of zonal mean MSF at 500 hPa within the latitudinal zone of 30°S–30°N (Oort and Yienger, 1996), respectively.

## 2.4 Phase synchronization of Walker cell and Hadley cell variability

To examine the phase synchronization of the large-scale circulation, we calculated the complex analytical signal ($\hat{T}(t)$) using

the Hilbert transform. The amplitude and phase of the PC time series can be obtained from $\hat{T}(t)$, based on a Cartesian to polar coordinate transform (e.g., Stein et al., 2014):

$$\alpha(t) = \sqrt{Re(\hat{T}(t))^2 + Im(\hat{T}(t))^2} \quad \text{and} \quad \Phi(t) = Arg\frac{Im(\hat{T}(t))}{Re(\hat{T}(t))}, \qquad (3)$$

where the generalized phase difference has the following expression:

$$\Delta\Phi(t) = \Phi_{WC}(t) - \Phi_{HC}(t), \qquad (4)$$

where $\Phi_{WC}(t)$ and $\Phi_{HC}(t)$ indicate the phase of the WC and HC PC time series respectively. If the absolute tendency of the phase difference (i.e., $|\frac{d(\Delta\Phi(t))}{dt}|$) is close to zero, then $\Phi_{WC}(t)$ and $\Phi_{HC}(t)$ are synchronized (Pikovsky et al., 2000). We use the following criterion for phases synchronization: phases synchronization occurs when the absolute tendency of a smoothed (i.e., 7-month running mean) phase difference of the WC and HC shift modes is less than 0.3 degrees/month. Strictly speaking, phase synchronization can only occur in nonlinear dynamic systems.

## 3 Result

### 3.1 Phase synchronized spatial shifts of Walker and Hadley circulations

First, we conduct an EOF analysis of the monthly MSF anomalies, obtained from atmospheric reanalysis data covering the period 1979-2017. In addition to their co-varying strength (characterized by EOF1 for WC (WC1) and EOF2 for HC (HC2) – shown in Supplementary Fig. S1) that has been discussed extensively in previous studies (Ma and Li, 2008;Bayr et al., 2014;Guo and Tan, 2018;Minobe, 2004), we find that the EOF2 of the WC (WC2) and the EOF1 of the HC (HC1) are related to each other (correlation coefficient CC ~ 0.49, Fig. 1a-c). Compared to the climatological circulation pattern (contours in Fig. 1a-b) of WC and HC, WC2 and HC1 describe zonal and meridional shifts occurring over the shared rising branch of the two circulations (shading in Fig. 1a-b) over the Western Pacific Warm Pool. These anomalies characterize an eastward shift of the WC and an equatorially asymmetric clockwise HC that is sometimes referred to as cross-equatorial anomalous HC (Supplementary Table S1 also clearly supports the co-varying zonal and meridional changes in WC and HC variability). Due to the spatio-temporal orthogonality of EOF modes, the co-varying spatial shifts of the WC and HC are uncorrelated to the strength changes of both circulations (e.g., CC between WC2 and WC strength index ~ 0.08; CC between HC1 and HC strength index ~ 0.02; see Supplementary Table S1).

Previous studies showed that the ENSO-related SST variability leads to the co-varying strength of WC and HC (i.e., Walker and Hadley strength modes) (e.g., Minobe, 2004;Guo and Tan, 2018). Here, we explore the relationship between ENSO and the co-varying spatial shifts of the WC and HC (Fig. 1c and e). No statistically significant linear correlation between the Principal Components (PCs) of HC1/WC2 and Niño3 anomalies can be found on interannual timescales (CC ~ 0.06 for WC2-Niño3; CC ~ 0.2 for HC1-Niño3). This indicates that the peak ENSO dynamics alone cannot explain the synchronized

spatial shifts in these two circulations. This stands in sharp contrast with the peak-phase ENSO-driven co-varying strength of the cells (CC ~ 0.91 for WC1-Niño3; CC ~ 0.87 for HC2-Niño3; see also Supplementary Fig. S1c) that has been highlighted previously. Despite the fact that there is no linear contemporaneous relationship between ENSO (Niño3 SST) and the co-

varying spatial circulation shifts, there is still a pronounced statistically significant relationship between WC2 and HC1 variability (see Fig. 1d; CC ~ 0.66 significant at 99% confidence level) in a 40 member MME of SST-forced AMIP experiments. This WC2-HC1 linkage breaks down when the SST-forced MME component is subtracted from the PCs of the individual AMIP runs (i.e., PC differences between individual AMIP simulations and the MME; Supplementary Fig. S2). This raises the question what type of SST pattern is responsible for the synchronized spatial shifts of WC and HC.

This discrepancy calls for two important aspects of ENSO to be considered: one is the peak-phase ENSO signal (which occurs in boreal winter); the other are the seasonally modulated characteristics of ENSO, i.e., the combination mode between ENSO and the Indo-Pacific warm pool SST annual cycle (C-mode; which a time evolution that peaks in boreal spring) (Stuecker et al., 2013;Stuecker et al., 2015). The C-mode arises from an amplitude modulation of the warm pool annual cycle (with a frequency of 1 year$^{-1}$) and the interannual ENSO signal (~ 1/2 - 1/7 year$^{-1}$ frequency). The C-mode plays a crucial

role in the phase transition of ENSO events as well as in bridging ENSO's impacts to the East Asian monsoon system (Stuecker et al., 2013;Stuecker et al., 2015). We emphasize that the meridionally asymmetric anomalous atmospheric C-mode circulation is mostly associated with strong El Niño events occurring in the eastern equatorial Pacific and less so for central Pacific El Niño events (McGregor et al., 2013) (e.g., CC ~ 0.55 for WC2- Niño3 during boreal spring; CC ~ 0.09 for WC2-Niño4). We hypothesize here that the seasonally evolving warm pool SST anomalies after the peak ENSO phase (i.e., C-mode) serve as a

pacemaker linking the phase-synchronized spatial shifts in WC and HC variability. The seasonal differences in relationship between WC2/HC1 and Niño3 indices support our hypothesis (see Supplementary Fig. S3). The WC2-regressed SSTA pattern (bottom in Fig. 1a) also bears resemblance with the previously suggested C-mode generated SST dipole pattern with anomalous warming in the Southern Indian Ocean (SIO) and anomalous cooling in the WNP (Stuecker et al., 2015;Zhang et al., 2016). This suggests that the Walker and Hadley shift mode variability is linked via nonlinear C-mode dynamics and not well reflected

by common ENSO indices (such as Niño3) that instead describe peak boreal winter ENSO variability and the associated Walker and Hadley strength modes (Supplementary Fig. S1).

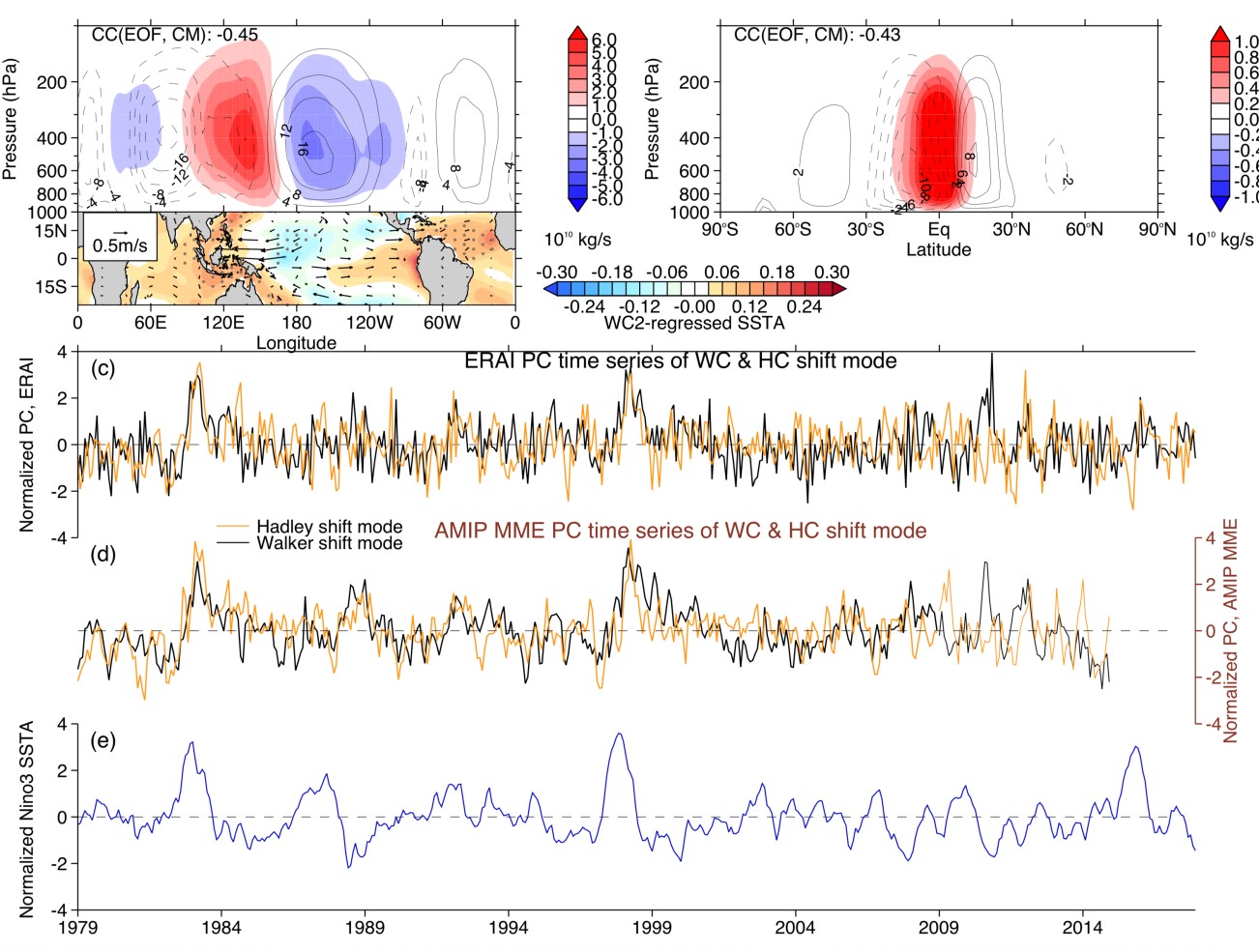

**Figure 1: Phase synchronized spatial shifts of the Walker and Hadley circulations.** (a-b) The dominant patterns of the shift modes (i.e., EOF2 for Walker and EOF1 for Hadley; denoted by shading) for (a) Walker circulation (WC) and (b) Hadley circulation (HC) variability. The WC and HC are identified as (a) tropical (5°S-5°N) averaged mass stream function (MSF) anomalies and (b) zonally averaged MSF anomalies, obtained from the monthly ERA interim (ERAI) data during the period 1979-2017. The contours show the climatological mean (CM) of the MSF for the circulations. The pattern correlation coefficients (CCs) between EOF and CM are also displayed in (a-b). The SST and 850hPa wind anomaly patterns regressed against the WC2 principal component (PC) are displayed as a bottom inset in (a). (c-d) The normalized PC time series of the shift modes (i.e., WC2 and HC1), derived from (c) ERAI and (d) the multi-model ensemble (MME) average for 40 AMIP models (1979-2008 for 19 AMIP5 and 1979-014 for 21 AMIP6 models). (e) The normalized Niño3 SST anomaly (ERSSTv5) time evolution.

## 3.2 SST-forced variability of Walker and Hadley circulations

In the previous section, we showed that the "Walker and Hadley shift modes" are coupled to each other, and we hypothesized an important role of the seasonally modulated dynamics of ENSO. The previously applied linear correlation

analysis is well suited to analyze the amplitude relationships, however, it is not sensitive to the seasonally modulated nonlinear coupling. Therefore, before focusing on the physical linkage between these shift modes, we further examine the co-varying

WC-HC shifts in the framework of phase synchronization and nonlinear coupling. Phase synchronization, which requires nonlinear dynamics, is characterized by cooperative and organized oscillatory behavior between two fluctuating systems. To study this process, we first calculate the generalized phase difference between WC2 and HC1 variabilities (i.e., $\Delta\Phi(t) = \Phi1 - \Phi2$), using the complex analytical signal of the respective principal components WC PC2 and HC PC1 (section 2.4). The analytical signal approach embeds a timeseries into a complex space by adding the original timeseries with the Hilbert

transform of this timeseries times the imaginary number (Pikovsky et al., 2000;Stein et al., 2014;Rosenblum et al., 1998;Rosenblum, 2000;Tass et al., 1998). The phase difference between WC2 PC and HC1 PC (Fig. 2a) shows periods of near constant phase difference. Constant phase differences, which occur more frequently than expected by randomness, are indicative of the emergence of *phase synchronization* between WC and HC shift modes. When the absolute tendency of the phase difference (i.e., $|d(\Delta\Phi(t))/dt|$) in our calculations is less than 0.3 degrees/month (Fig. 2a), we refer to the period as a

phase-synchronized period. During these periods the phases of two signals are not drifting apart randomly but are bounded by the underlying nonlinear interactions.

We next examine the question whether the synchronization of "Walker and Hadley shift modes" can in principle be driven by random atmospheric stochastic variability, or if SST-forced variability is a prerequisite. The SST-forced AMIP MME shows more frequent phase-synchronized (PSYN) months than the observations (Fig. 2a) and most CMIP5 models

(Supplementary Fig. S4), exhibiting the effects of both SST forcing and atmospheric noise: There are four long-lasting (greater than 6 months) PSYN periods in the observations but eight such events in the AMIP MME. The WC-HC phase synchronization is particularly prominent during the two extreme El Niño events (1982-1893 and 1997-1998) that have much longer PSYN periods (exceeding 1-year) in both observations and the AMIP MME. An interesting point is that the WC-HC phase difference shows a decadal change after 2000 in both observations and the AMIP MME. This may be a consequence of the Pacific WC

intensification which can be linked to the concurrent tropical Atlantic warming and tropical eastern Pacific cooling (England et al., 2014;McGregor et al., 2014; associated with multi-decadal climate variability in both the Atlantic and Pacific) and/or zonal shifts in the dominant ENSO SST pattern (Sohn et al., 2013). Further exploration of the underlying physical mechanisms is, however, beyond the scope of this study.

To further explore the relative effects of random atmospheric noise and SST-forced variability on the WC-HC

synchronization, we generate pseudo-PCs by randomly shuffling the PCs for both the observations and the AMIP MME and repeating this 10 times. A comparison of the phase difference tendencies using the original PCs (pink dots in Fig. 2b, c)) and of the pseudo-PCs (sky blue dots) reveals that during the evolution of El Niño events (Niño3 > 1.5 standard deviations (SDs); denoted by gray shading), the phase synchronization (i.e., $|d(\Delta\Phi(t))/dt| < 0.3$) of the original PCs is clearly distinguishable from the pseudo-PCs that have randomized phases: The frequency for phase synchronization (i.e., the ratio of the PSYN

number compared to the number in the gray shaded area) is ~ 83% for the observations and 96% for the AMIP MME, whereas the ratio for the pseudo-PCs (sky blue dots) is only ~33% in the observations and 39% in the AMIP MME (Figs. 2b and 2c).

During the evolution of La Niña events (Niño3 < -1.5SD), the phase synchronization is indistinguishable from random variability, hinting also to an important nonlinearity in the atmospheric response to tropical Pacific SST forcing. The PSYN probability from pseudo-PCs in case of extreme El Niño events is similar to the probability for the entire dataset (~ 35%).

Consequently, this result indicates that even though the temporal correlation between all month Niño3.4 SSTA and PC2 WC and PC1 HC is marginal, eastern equatorial Pacific SST forcing still leads to an increased probability for WC-HC phase synchronization, during the El Niño event evolution . This discrepancy can be explained by the fact that correlations emphasize amplitude relationships and are less sensitive to seasonally modulated coupling which is induced by the phase-synchronization mechanism.


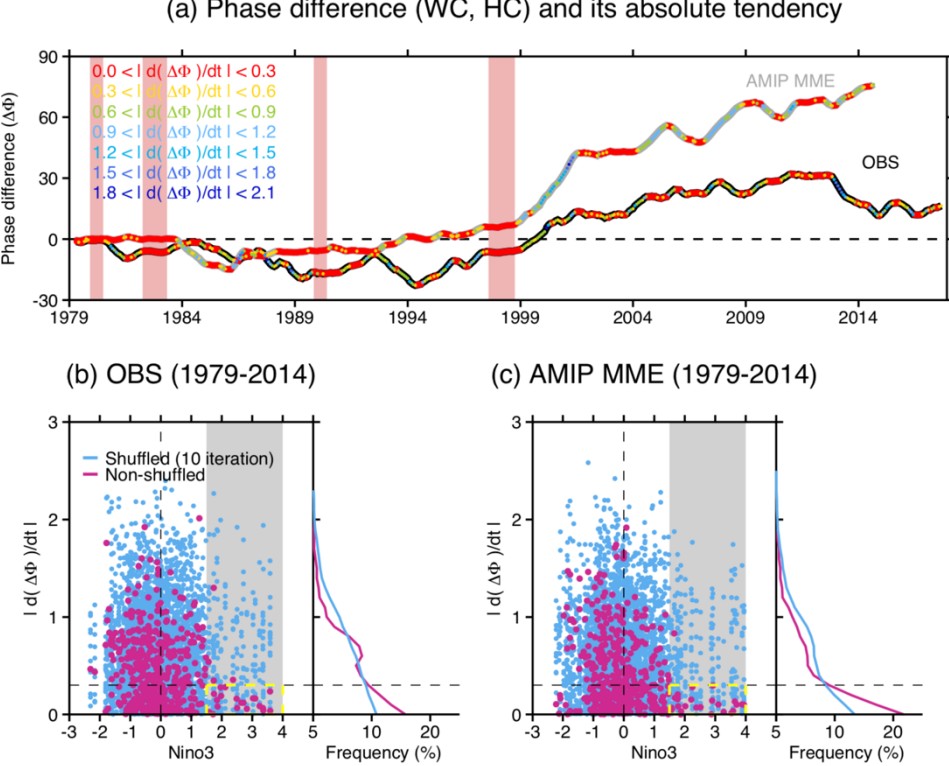

**Figure 2: SST-forced phase synchronized spatial shifts of the HC and WC.** (a) The smoothed (i.e, 7-month running mean) phase difference between PC time series of WC2 and HC1 (position of dots) and its absolute tendency (color of dots), obtained from the ERAI observational data (1979-2017; black) and AMIP MME (1979-2014; gray). Here, the phase is calculated from the analytical signal of the 215 PCs using the Hilbert transform (see Method section for detail). The phase synchronized (PSYN) months are measured by the criterion that the absolute tendency of phase difference is less than 0.3 °/month. The red vertically shaded zone indicates PSYN periods that are identical for both observations and the AMIP MME. (b-c) The absolute tendency of WC-HC phase difference as a function of Niño3 SST anomaly (pink), obtained from the (b) observations and (c) AMIP MME. Sky blue dots indicate tendencies calculated from the randomly shuffled PCs. The frequency histogram of the absolute tendency of the WC-HC phase difference is also vertically plotted in each right inset.


Our analysis has revealed that the "Walker and Hadley shift modes" are more connected to the seasonally modulated dynamics of ENSO (i.e., the C-mode) rather than the peak-phase ENSO amplitude signal. Figure 3a clearly shows the statistically significant co-variability between observed time series of WC-HC PCs and the C-mode (which is defined here as

$Niño3 \times cos(\omega_a(t) - \varphi)$; where $\omega_a(t)$ is angular frequency of the annual cycle and and $\varphi$ is a one-month phase shift; (Stuecker et al., 2013)) (CC ~ 0.61 and 0.58 on inter-annual timescales). To further test the association with the C-mode, we also consider an SST index representing the integrated effect of C-mode dynamics (See also discussion in (Zhang et al., 2016)), which is calculated from the SST gradient between the SIO [20°S-5°S, 90°E-120°E] and the WNP [0°-15°N, 140°E-170°E] (i.e., SSTgrad; ΔSST(SIO, WNP)), as shown by the WC2-regressed SST pattern in Fig. 1a. We can see a strong seasonal characteristics in both SST indices (Niño3, C-mode, and SSTgrad) as well as in the shift modes (Figs. 3b-3g). The common Niño3 index peaks in boreal winter (i.e., November-December-January; NDJ; blue dots) in both observations and the AMIP MME. Different from this winter-maximum El Niño variability, the SSTgrad (or C-mode) variability reaches its maximum during boreal spring (i.e., February-March-April; FMA; red dots). The difference in seasonality, which is also clearly seen in Supplementary Fig. S5, can explain the very low correlation coefficient between the common Niño3 index and the shift modes. Despite of this insignificant correlation, during the boreal spring, warmer Niño3 anomalies tend to be associated with enhanced shift mode variability, however, the opposite is not true for cold Niño3 anomalies (i.e., La Niña). This is consistent with the previous conclusion that the WC-HC phase-synchronization only occurs during extreme El Niño events, but not during extreme La Niña events. This springtime seasonal preference is evident for the SSTgrad (or C-mode) index, resulting in a statistically significant relationship between SSTgrad (or C-mode) and WC-HC coupling in both observations and the AMIP MME (Figs. 3d and 3g). This again indicates the important impact of springtime SSTgrad anomalies and C-mode dynamics on the phase synchronization of WC-HC.

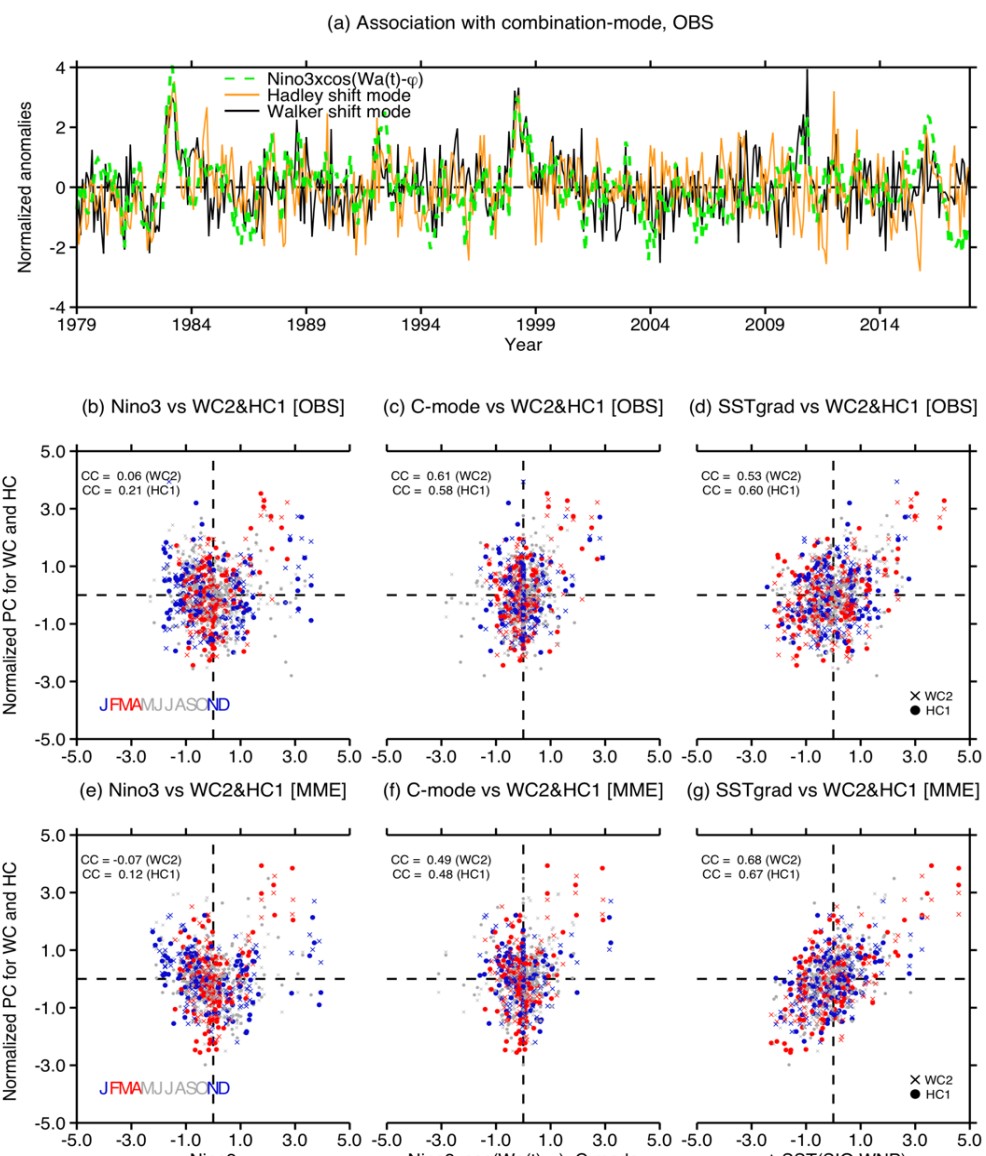

**Figure 3: Seasonal dependency of phase synchronized spatial shifts of the HC and WC.** (a) The normalized PC time series of the shift modes (i.e., WC2 & HC1) and the combination mode (Stuecker et al., 2013; dashed green line). (b-g) Scatter plot of normalized time series for SST anomalies (x-axis) versus PCs of WC2 (x symbol) and HC1 (y-axis), obtained from the observations (1979-2017; b-d) and the AMIP MME (1979-2014; e-g): (b, e) Niño3, (c, f) combination mode, and (d, g) SST gradient between Southern Indian Ocean [20°S-5°S, 90°E-120°E] and western North Pacific [0°-15°N, 140°E-170°E] (SSTgrad). Each season is displayed in different colors.

## 3.3 Global pattern and impact of phase synchronized spatial shifts

Our results suggest that the spring SSTgrad anomalies that occur after the El Niño winter peak phase (Stuecker et al., 2015;Zhang et al., 2016) play an important role for the phase synchronization of WC and HC shift modes. We further examine the physical mechanism for the WC-HC phase synchronized spatial shifts by exploring the global climate patterns that occur during springtime phase-synchronized periods in WC and HC variability (Fig. 4; composite when $|d(\Delta\Phi(t))/dt| < 0.3$ °/month and FMA[1] Niño3 > 1.5SD). A key feature is that the center of low-level divergence (upper-level convergence) and drying is located over the off-equatorial WNP region, whereas the center of convergence and moistening is situated in the off-equatorial southern central Pacific (corresponding to the C-mode pattern discussed in previous studies (Stuecker et al., 2015;Stuecker et al., 2013)). This equatorially asymmetric circulation pattern is considerably different from the equatorially symmetric pattern derived from the wintertime PSYN composite (see Supplementary Fig. S6) and even the springtime non-PSYN composite (Supplementary Fig. S7). The meridionally asymmetric WC can be linked to the HC, along with cross-equatorial northwesterly flow. A general agreement between the observation-based and AMIP MME-based patterns also reflects that the SST-forced variability acts to modulate the phase synchronization of the WC-HC shift modes.

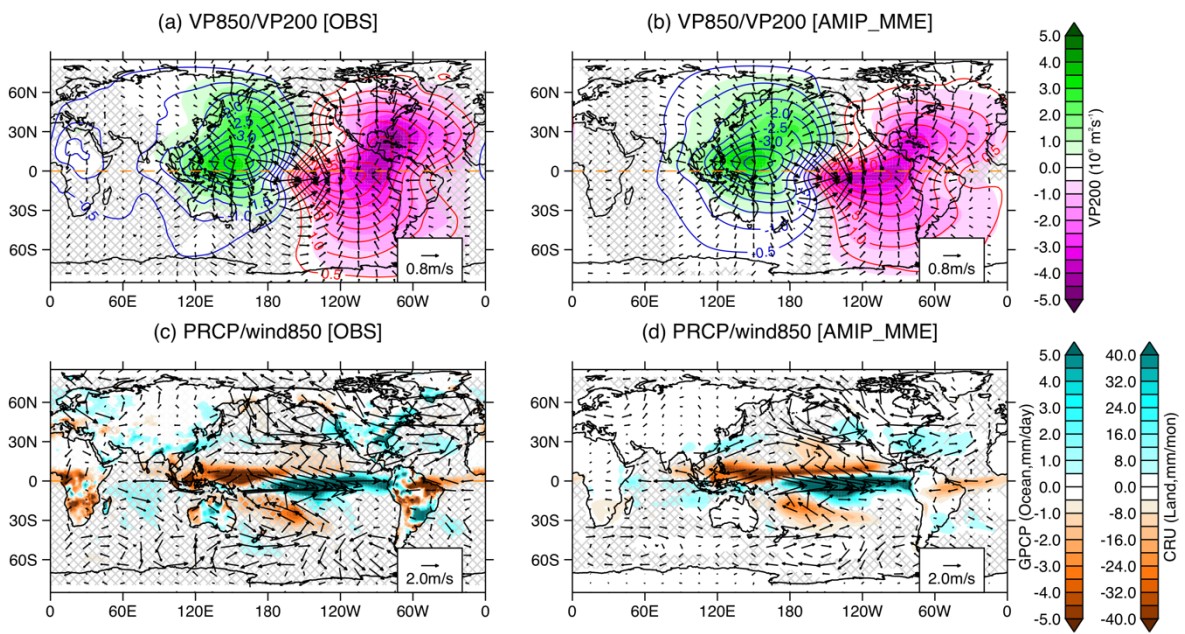

**Figure 4: Global pattern of phase synchronized spatial shifts of HC and WC.** Composite anomalies during PSYN months (i.e., the absolute tendency of phase difference is less than 0.3) with February-March-April (FMA[1]) extreme El Niño (FMA[1] Niño3 > 1.5SD), obtained from the observations (left column) and the AMIP MME (right column): (a, b) 850 hPa velocity potential (VP850; contours) and 200 hPa velocity potential (VP200; shading) anomaly; (c-d) precipitation (GPCP for ocean and CRU for land; shading) and 850hPa wind (vector) anomaly. The hatching shows the area where the difference is statistically insignificant at the 99% confidence level.

We emphasize that the simulated ENSO SST anomaly amplitudes in coupled models of the CMIP5 are correlated with the probability of phase synchronization between the WC and HC shift modes (Supplementary Fig. S4). The models

exhibiting large ENSO variability can initiate strong teleconnections through the atmospheric bridge process, thus likely leading to the phase-synchronized zonal and meridional shifts of the WC and HC, respectively (see Supplementary Fig. S8). An underlying cause could be the strong meridional asymmetry of the warm pool climatological SST and zonal winds (Supplementary Fig. S9) and C-mode associated extreme shifts of the South Pacific Convergence Zone (SPCZ) shifts (McGregor et al., 2012). We further hypothesize that stronger phase synchronization of the WC-HC shift modes can generate larger global precipitation responses, regardless of the ENSO amplitude. To illustrate this, we use the individual 40 AMIP simulations that have identical ENSO SST amplitude prescribed as their boundary forcing but exhibit different strengths of WC2-HC1 phase synchronization. The strong PSYN models and weak PSYN models are thus classified as a function of their strength of WC2-HC1 phase synchronization, where this strength is measured by the (inter-annual) correlation coefficient between WC2 and HC1 PC variations in the individual AMIPs (see Supplementary Tables S2 and S3 for chosen model groups).

By comparing the strong El Niño (i.e., 1982/83 and 1997/1998) composite averaged for eight strong PSYN models with that averaged for eight weak PSYN models (Fig. 5), we see that the models with a strong coupling between the HC and WC shift modes generate a prominent springtime (FMA) asymmetry between WNP divergence and southern central Pacific convergence. In particular, the geographical positions of minimum low-level divergence and maximum low-level convergence exhibit a clear contrast between the strong PSYN models and weak PSYN models (see pink and cyan star symbols): the NW-SE oriented pattern is predominant for the strong PSYN models, but it is less pronounced for the weak models. Accordingly, the increased precipitation anomalies in the extratropical regions (e.g., East Asia and South America) are larger for the strong PSYN models compared to the weak PSYN models (Figs. 5b and 5d). These differences in anomalous precipitation and circulation centers are statistically significant during boreal spring but not during boreal winter. We also emphasize that these differences in precipitation are statistically insignificant for the tropical regions, indicating that the extratropical precipitation response is not primarily a result from an enhanced tropical convective activity. Differences in model physics may generate a strengthened WC-HC synchronization (i.e., NW-SE skewed asymmetric circulation pattern changes), further intensifying the post-ENSO impact on extratropical precipitation variability. This reflects the pronounced impact of WC-HC phase-synchronized spatial shifts on extratropical precipitation variability, regardless of the ENSO amplitude.

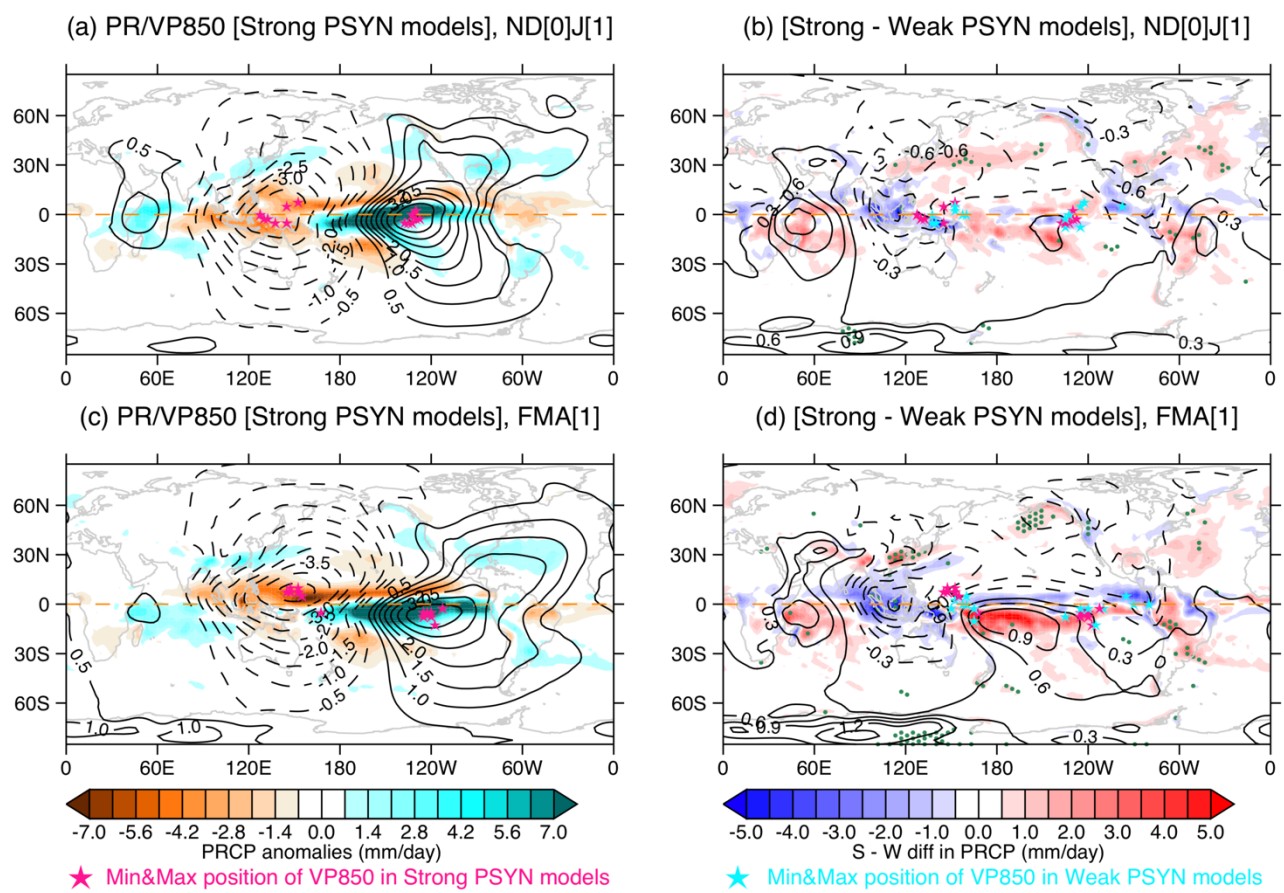

**Figure 5: Impact of phase synchronized spatial shifts of the HC and WC on global precipitation.** (a) The November-December[0]-January[1] extreme El Niño (i.e., 1982/1983 and 1997/1998) composite anomaly of precipitation (shading) and 850hPa velocity potential (VP850; contours) obtained from the 8 strong PSYN models in the AMIP simulations and (b) the difference between 8 strong PSYN models and 8 weak PSYN models. (c-d) Same as (a-b), but for February-March-April (FMA[1]). The green dots in (b, d) indicate the area where the precipitation difference is statistically significant at the 99% confidence level. The pink and cyan star symbols indicate the positions of maximum and minimum VP850 anomalies for the strong PSYN and weak PSYN models. Here, the strong PSYN and weak PSYN model groups are categorized according to inter-annual correlation coefficients between WC2 and HC1 within 19 AMIP5 and 21 AMIP6 models, respectively (see Supplementary Tables 2 and 3).

## 4 Discussion and Conclusion

Focusing on the zonal and meridional displacements of two major atmospheric circulation cells, using observational datasets and the AMIP MME simulations, we examined the intricate coupling between the WC and HC. In addition to a well-known ENSO driven coupling of WC and HC variability amplitude, our study shows that these two circulations can also shift their positions in a synchronized manner, albeit in a more subtle manner. Figure 6 illustrates the important difference between the

strength modes (WC1&HC2) and the shift modes (WC2&HC1). The ENSO-driven strength mode in its winter maximum is characterized by an equatorially symmetric Pacific cell pattern between western Pacific divergence and central Pacific convergence, leading to weakening WC and strengthening HC strengths (Fig. 6a). In contrast, our analysis revealed that the seasonally evolving springtime warm pool SSTs in combination with post-peak phase El Niño SST anomalies can produce a meridionally asymmetric WC and anomalous cross-equatorial flow (from off-equatorial WNP divergence zone toward the off-equatorial southern Ocean of central Pacific and Indian Ocean convergence zone), thus connecting the zonal WC and meridional HC (Fig. 6b). This feature is reminiscent of the atmospheric ENSO combination mode. Coherent WC-HC shifts have pronounced influences on precipitation, as well as wind and sea-level patterns in both tropical and extratropical regions.

An important finding of our study is that the phase-synchronization is only present for extreme El Niño events and not for La Niña, indicating an important nonlinearity of the tropical climate system. In addition, potential asymmetric heat capacitor effects of both the Indian and Atlantic Oceans, that tend to be more pronounced during El Niño compared to La Niña (Ohba and Watanabe, 2012;An and Kim, 2018) may further affect the phase synchronization properties discussed here. We found that the meridionally asymmetric response for phase synchronization of the WC-HC shift mode is quite different from the meridionally symmetric response associated with the strength-related WC-HC modes (Supplementary Figs. S10 and S11). Our analysis also highlights the fact that the analysis of phase-relationships (Rosenblum and Pikovsky, 2003) between modes of climate variability can reveal important new insights into their physical coupling mechanisms, that are less apparent in amplitude space (as described for instance by temporal correlation coefficients).

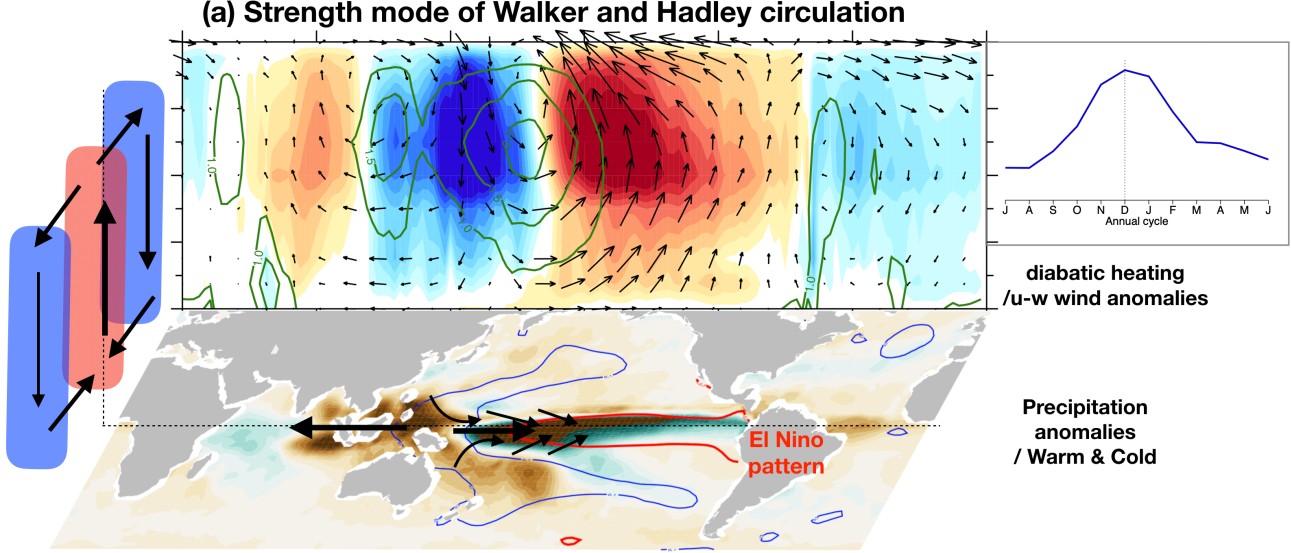

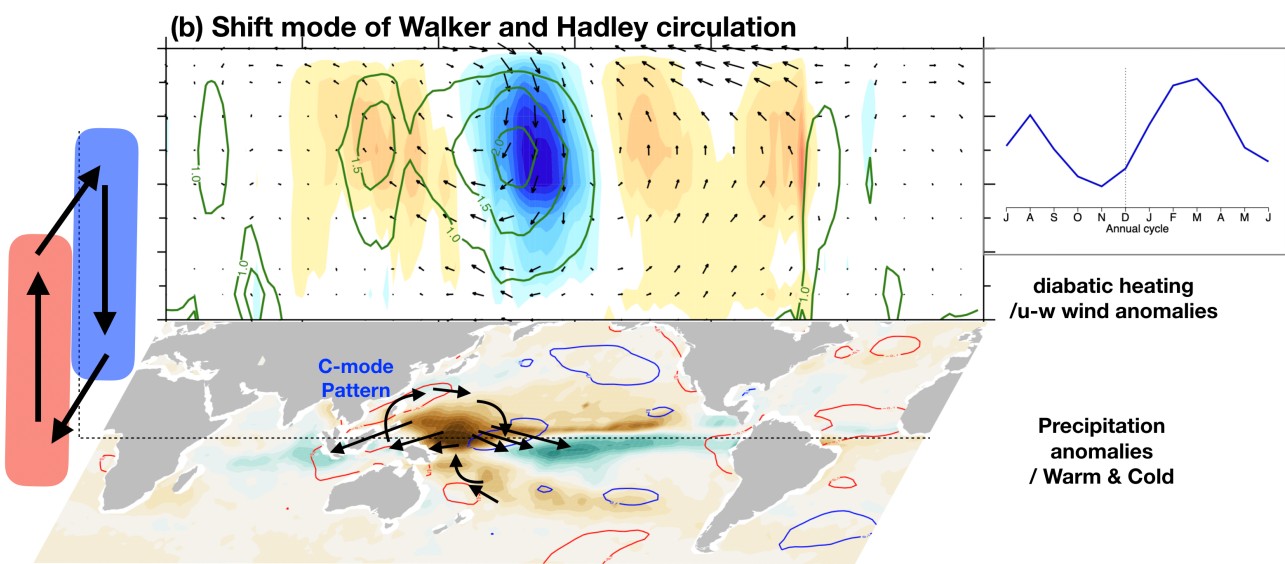

**Figure 6: Schematic diagram showing the strength and spatial shift of Walker and Hadley circulations.** (a) Walker and Hadley circulation strength mode (WC1&HC2) and (b) shift mode (WC2&HC1) on inter-annual timescale. The upper-panel shows the longitude-height structure averaged over tropics [15ºS-15ºN] for the diabatic heating anomalies (shading), climatological diabatic heating (green contours), and u-w wind anomalies (vectors). The bottom horizontal map shows the precipitation anomalies (shading) and warm/cold SST anomalies (red/blue contours). The schematic on the left indicates the anomalous Hadley circulation. All plots are based on the regressed

anomalies against the Walker circulation variability (WC1 in (a) and WC2 in (b)). The upper right panels show the annual cycle of circulation variability.



## Code and Data availability

Observation data for this research are available at (1) ECMWF website (https://www.ecmwf.int/en/forecasts/dataset/ecmwf-reanalysis-interim) for ERA-Interim reanalysis data, (2) NCDC ([https://www.ncdc.noaa.gov/data-access/marineocean-data/extended-reconstructed-sea-surface-temperature-ersst-v5](https://www.ncdc.noaa.gov/data-access/marineocean-data/extended-reconstructed-sea-surface-temperature-ersst-v5)) for ERSST v5, and (3) ESRL
([https://www.esrl.noaa.gov/psd/data/](https://www.esrl.noaa.gov/psd/data/) gridded/data.gpcp.html) for GPCP v2.3, and (4) the website (https://crudata.uea.ac.uk/ cru/data/hrg/ cru_ts_4.03/) for CRU TS4.03.The model data for both AMIP and CMIP are also available at a publicly accessible CMIP5 website (https://esgf-node.llnl.gov/search/cmip5/) and CMIP6 website (https://esgf-node.llnl.gov/search/cmip6/).

**Author Contributions**

AT and K-SY designed the study. K-SY wrote the initial manuscript draft and produced all figures. K-SY, AT, and MFS contributed to the interpretation of the results and to the improvement of the manuscript.

**Acknowledgements**

This study was supported by the Institute for Basic Science (project code IBS-R028-D1). This is IPRC publication X and SOEST contribution Y.

**Competing Interests**

The authors declare that they have no competing financial interests.

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
