# Peer review of "Synchronized spatial shifts of Hadley and Walker circulations"

_Earth System Dynamics, 2020_

## Referee Comment (RC1) · Anonymous Referee #1 · 26 Oct 2020

General Comments

"Synchronized spatial shifts of Hadley and Walker Circulations", by Yun et al., is an interesting analysis of the phase and amplitude of the time series of modes of Hadley and Walker cell circulations. However, the closest thing to a central hypothesis in the paper, "...that the seasonally evolving warm pool SST anomalies after the peak ENSO phase serve as a pacemaker linking the phase-synchronized special shifts in the WC and HC variability..." is well-established by the analyses presented in the paper, but it feels more like a summary of some of the results than a hypothesis to be tested by the analysis. Rather than aiming for a specific objective or to test a hypothesis, this paper seems more an exploration of modes of Hadley cell-related and Walker cell-related variability. As a consequence, the paper feels at times like a collection of figure panels

with a loose narrative joining them. Indeed, the discussion of the figures can be as brief as the captions themselves. Indeed, I get the feeling that there is more to discuss when looking at these figures than is discussed in the text. At other times, I feel like too much is being made of a blob of color in one figure or other. But the analysis of the synchronization of the Walker and Hadley modes (or at least the 2nd Walker Cell mode and the 1st Hadley cell mode) is interesting, and the authors make a pretty convincing case that the synchronization of these modes peaks after an El Niño – hence the above pacemaker analogy – rather than through random chance. I think the analysis of phase and amplitude was a highlight of the paper. The paper concludes with a brief discussion of the importance of synchronized shifts in explaining precipitation variability in East Asia and the South Atlantic–a discussion I found wanting. All in all, though, this was an interesting analysis of the connections between Hadley and Walker cell shifts, and this manuscript has the potential to be a great place to start for readers of ESD wanting to examine the phenomenology of such synchronization more closely.

Specific Comments

- Since the synchronized circulation induced by the SST anomalies in the warm pool is a diagonal (i.e. oriented NW-to-SH) overturning circulation, how useful do you think the Hadley Cell-Walker Cell framework is in characterizing it?

- Why use forced AMIP, CMIP, and Historical obs if you're focusing on variability? Why not use a larger dataset, like the CESM LE?

- HC1, HC2, WC2, WC2—can you be clear about what you think each mode physically corresponds to?

- Some figures have seemingly redundant panels; Figure 2, for example, has b, c, d, and e, but they're not discussed much. It seems like b, and c show the same message as d and e. If these panels tell an interesting story, please share it. Otherwise, cut down on the number of panels to be commensurate with the text.

- How different is the global pattern of velocity potential and precip phase-synchronized HC and WC compared to other just-after-ENSO years?

- I mentioned I found the precipitation discussion wanting. Part of that may be because of weaknesses of Figure 5. The first four panels of Figure 5 all look almost the same, with very slight differences in shading between the SA and EA boxes in some panels. If you're discussing the difference, maybe show the anomalies of some panels with respect to the others—again, they all just look like Walker-like ink splots, and the differences inmagnitude aren't all that clear. Also, panel 5e shows the box-and-whisker plots for. . .strong AND weak points? The Strong and Weak points themselves are superimposed using color (which is muddied a bit in print). So this panel tells several different, related stories—more than the paper tells. And while there is a difference in the precipitation anomalies between the strong PSYN models and the weak PSYN models, it seems just as likely to me that some model physics are causing the hydrological differences and the synchronization, but that the two aren't necessarily directly related. Perhaps diabatic heating and cooling by the altered hydrology of one model, owing to its physics, produces the synchronization, rather than (as your paper posits) the other way around.

Technical Corrections

- Line 29: I would add Kris Karnauskas's 3D Hadley circulation paper to this list: Karnauskas, K.B., Ummenhofer, C.C. On the dynamics of the Hadley circulation and subtropical drying. Clim Dyn 42, 2259–2269 (2014). https://doi.org/10.1007/s00382-014-2129-1

- Line 35–36: I would change, ". . .changes during. . ." to ". . .changes, generally during. . ."

- Line 40: this question-in-a-sentence could have its syntax improved.

- Line 114: ". . .there no. . ." should be ". . .there is no. . ."

- Line 125: a bit more intro to the C-mode would be nice.

- Figure 1a, b: The font inside the figures gets small enough as to render it very hard to read in print.

- Figure 1c, d: based on the panel titles, "Interannual CC of...", I thought the plot was showing a running (windowed?) cross-correlation between WC2, HC1, and...something else. After flipping back and forth and checking the y-axis, I realized that it was showing the PC time series themselves, thereby _illustrating_ the correlation.

- Figure 2 (line 184): be clear that the COLOR of the dots specifies the absolute derivative, while the position of the dots illustrates the phase difference.

- Figure 3: the green and blue dots on red are very muddy on my printed page—and I'm not even colorblind!

---

## Referee Comment (RC2) · Anonymous Referee #2 · 3 Nov 2020

This paper shows that warm pool SST anomalies in the decaying El Niño event generate a meridionally asymmetric Walker circulation response, which couples the zonal and meridional atmospheric overturning circulations. I think their results are overall novel and reasonable, and I would be happy to see this work on Earth System Dynamics after some minor revisions, particularly regarding discussions. Specific comments are as follows.

1) The authors focus on NINO 3 regions to explore the relationship between SST anomalies and Walker circulation. Recently, Central Pacific El Nino events tend to increase and also some papers suggested the increase of CP El Nino in a warm climate. I think that the relationship may be changed if we concentrate on the NINO4 region. The authors need to discuss the sensitivity of the NINO region on the relationship somewhere. 2) In figure 1, the temporal evolution of normalized PC from WC or HC almost coincides with the NINO3 index from 1979-2000. However, the relationship between the two indexes seems to be weakened after 2000. I would know possible reasons. I think that many reasons may be discussed – ENSO diversity, mult-idecadal variability (IPO or AMO), and even global warming. 3) In figure 2, it is well known that the AMIP run tends to overestimate the atmospheric response given SST forcing. How much does the strength or duration length of phase synchronization may be changed in a coupled model? 4) In Figures 4 and 5, I would see the circulation pattern in the upper troposphere (200 hPa). If the authors think the upper-level circulation change is not relevant to this study, please mention the reason in the main text. 5) In figure 4, the 95 % significance level may be too low to show a strong shift of HC and WC. Why don't you use 99% or other higher criteria?

Please also note the supplement to this comment:
https://esd.copernicus.org/preprints/esd-2020-70/esd-2020-70-RC2-supplement.pdf
* * *

---

## Author Comment (AC1) · 2 Dec 2020

[Research Article # ESD]: "Synchronized spatial shifts of Hadley and Walker circulations" by Kyung-Sook Yun, Axel Timmermann, Malte F. Stuecker

We thank the reviewers for their constructive and helpful comments. We carefully revised the manuscript "Synchronized spatial shifts of Hadley and Walker circulations" and provide a point by point reply to the individual comments below.

**Reply to the comments of Reviewer #1**

**General Remarks:**
*"Synchronized spatial shifts of Hadley and Walker Circulations", by Yun et al., is an interesting analysis of the phase and amplitude of the time series of modes of Hadley and Walker cell circulations. However, the closest thing to a central hypothesis in the paper, ". . .that the seasonally evolving warm pool SST anomalies after the peak ENSO phase serve as a pacemaker linking the phase-synchronized special shifts in the WC and HC variability. . ." is well-established by the analyses presented in the paper, but it feels more like a summary of some of the results than a hypothesis to be tested by the analysis. Rather than aiming for a specific objective or to test a hypothesis, this paper seems more an exploration of modes of Hadley cell-related and Walker cell-related variability. As a consequence, the paper feels at times like a collection of figure panels with a loose narrative joining them. Indeed, the discussion of the figures can be as brief as the captions themselves. Indeed, I get the feeling that there is more to discuss when looking at these figures than is discussed in the text. At other times, I feel like too much is being made of a blob of color in one figure or other. But the analysis of the synchronization of the Walker and Hadley modes (or at least the 2nd Walker Cell mode and the 1st Hadley cell mode) is interesting, and the authors make a pretty convincing case that the synchronization of these modes peaks after an El Niño – hence the above pacemaker analogy – rather than through random chance. I think the analysis of phase and amplitude was a highlight of the paper. The paper concludes with a brief discussion of the importance of synchronized shifts in explaining precipitation variability in East Asia and the South Atlantic–a discussion I found wanting. All in all, though, this was an interesting analysis of the connections between Hadley and Walker cell shifts, and this manuscript has the potential to be a great place to start for readers of ESD wanting to examine the phenomenology of such synchronization more closely.*

**[Ans]** We are grateful to the reviewer for very constructive comments which were helpful to improve this work. We modified the entire manuscript to give more narrative and to improve the Figures. The manuscript has been largely revised according to the reviewer's comments as listed below.

**Specific comments:**
*Q. 1. Since the synchronized circulation induced by the SST anomalies in the warm pool is a diagonal (i.e. oriented NW-to-SH) overturning circulation, how useful do you think the Hadley Cell-Walker Cell framework is in characterizing it?*

**[Ans]** To place our result of synchronized WC-HC variability in the context of the existing extensive literature on the WC and HC respectively, we here applied a simple EOF-based way to define the Walker and Hadley cells. We think it is an advantage of this analysis

that the NW-to-SH oriented WC-HC synchronization emerges naturally, even though our EOF analysis is conducted in N-S and E-W space explicitly. We believe that the simple EOF-based framework can capture the diagonal WC-HC shifts in both zonal and meridional directions and that the clear phase-synchronized signals between zonal mode and meridional mode support the usefulness of applied framework.

*Q. 2. Why use forced AMIP, CMIP, and Historical obs if you're focusing on variability? Why not use a larger dataset, like the CESM LE?*

**[Ans]** The AMIP run has the merit of having the same observed SST forcing. This strength can provide a fair comparison of the SST-forced atmospheric responses between observations and model simulations. Therefore, we used the AMIP MME to represent the SST-forced atmospheric response only and the individual AMIP models to show a mixed signal of the SST-forced variability and unforced atmospheric noise. In contrast, in the CMIP and large-ensemble simulations it is difficult to separate the respective contribution between SST-forced and SST-unforced variabilities, because of different spatio-temporally evolving SST anomaly patterns . We also used the CMIP models to examine the impact of ENSO variability amplitude on the synchronization of Walker and Hadley circulation. The use of large ensembles is beyond the scope of this study. We added some explanation for the dataset used in the revised text [line 65-70].

*Q. 3. HC1, HC2, WC1, WC2—ˇcan you be clear about what you think each mode physically corresponds to?*

**[Ans]** To clearly explain the physical linkage between WC1&HC2 and between WC2&HC1, we illustrate a schematic diagram showing their physical links (Ref_Fig.1). During El Niño events, there is an weakening of WC and an eastward shift of the rising branch over the Maritime Continent (MC), and a strengthening of HC and an enhanced upward motion in the equatorial region (WC1&HC2; Fig. Ref_1a). Although the WC shows a mixed feature of strength and spatial shift, both of the WC and HC variability is significantly linked to the strength changes of both cells (e.g., interannual correlation coefficient CC between the WC1 variability and WC strength index ~ 0.84; CC between the HC2 variability and HC strength index ~ 0.82). However, Due to the spatio-temporal orthogonality of EOF modes, the shift modes are uncorrelated to the strength indices (e.g., CC between WC2 and WC strength index ~ 0.08; CC between HC1 and HC strength index ~ 0.02). Here, the strength indices were respectively calculated by the maximum value of tropical mean [5°S-5°N] MSF at 500 hPa and by the maximum of zonal mean MSF at 500 hPa within the latitudinal zone of 30°S–30°N (Oort and Yienger, 1996). In contrast, the shift mode (i.e., WC2&HC1) consists of a meridionally asymmetric anomalous WC and anomalous cross-equatorial wind from off-equatorial WNP divergence zone towards the off-equatorial southern central Pacific convergence zone, thus indicating a zonal shift in WC and a poleward shift in HC. We added Ref_Fig. 1 (as new Fig. 6) and more explanation for the physical relationship between WC and HC in the revised text [line 118-120, 314-321 in the main text and page 2 line 11-16 in Supplementary text].

[Figure]

[Figure]

**Ref_Fig. 1 Schematic diagram showing the strength and shift of Walker and Hadley circulations.** (a) Walker and Hadley circulation strength mode (WC1&HC2) and (b) shift mode (WC2&HC1) on inter-annual timescale. The upper-panel shows the longitude-height structure averaged over tropics [15ºS-15ºN] for the diabatic heating anomalies (shading), climatological diabatic heating (green line), and u-w wind anomalies (vector). The bottom horizontal map indicates the precipitation anomalies (shading) and warm/cold SST anomalies (red/blue lines). The latitude-height structure is also plotted in red/blue cylinder (warm/cold diabatic heating) and v-w wind vector. All plots are based on the regression anomalies against the Walker circulation variability (WC1 in (a) and WC2 in (b)). The upper right panels show the annual cycle of circulation variability.

*Q. 4. Some figures have seemingly redundant panels; Figure 2, for example, has b, c, d, and e, but they're not discussed much. It seems like b, and c show the same message as d and e. If these panels tell an interesting story, please share it. Otherwise, cut down on the number of panels to be commensurate with the text.*

      **[Ans]** We cut down the panel d and e in Fig. 2. The redundancy in other Figures was also double-checked.

*Q. 5. How different is the global pattern of velocity potential and precip phase-synchronized HC and WC compared to other just-after-ENSO years?*

**[Ans]** Ref_Fig. 2 shows the VP850 and precipitation anomalies during non-synchronized ENSO event (shading) and the difference between phase-synchronized events and non-synchronized ENSO events (contours) from observations and AMIP MME. The non-synchronized ENSO events show the zonal contrast pattern between the anomalous equatorial MC divergence and central Pacific convergence, whereas the phase-synchronized ENSO events present the NW-SE tilted circulation pattern. This major difference is statistically significant at the 99% confidence level (hatched area). We added the explanation in the revised text [line 259-260].

[Figure]

**Ref_Fig. 2 Global pattern of phase synchronized spatial shifts of HC and WC.** Composite anomalies during non-PSYN months (i.e., the absolute tendency of phase difference is greater than 0.3) with February-March-April (FMA[1]) El Niño (FMA[1] Niño3 > 0.5SD) and the difference between non-PSYN months and PSYN months (contour), obtained from the observations (left column) and the AMIP MME (right column): (a, b) 850 hPa velocity potential (VP850) and (c-d) precipitation anomaly. The hatching shows the area where the difference is statistically significant at the 99% confidence level.

*Q. 6. I mentioned I found the precipitation discussion wanting. Part of that may be because of weaknesses of Figure 5. The first four panels of Figure 5 all look almost the same, with very slight differences in shading between the SA and EA boxes in some panels. If you're discussing the difference, maybe show the anomalies of some panels with respect to the othersâ ̆Tˇagain, they all just look like Walker-like ink splots, and the differences in magnitude aren't all that clear. Also, panel 5e shows the box-and-whisker plots for strong AND weak points? The Strong and Weak points themselves are superimposed using color (which is muddied a bit in print). So this panel tells several different, related stories â ̆Tˇmore than the paper tells. And while there is a difference in the precipitation anomalies between the strong PSYN models and the weak PSYN models, it seems just as likely to me that some model physics are causing the hydrological differences and the synchronization, but that the two aren't necessarily directly related. Perhaps diabatic heating and cooling by the altered hydrology of one model, owing to its physics, produces the synchronization, rather than (as your paper posits) the other way around.*

**[Ans]** To emphasize the PSYN impact on precipitation, we replotted Fig. 5 (see Ref_Fig. 3) and removed the box-and-whisker plots. By comparing the maximum and minimum positions of VP850 between strong PSYN and weak PSYN models, we can clearly

see that the models with a strong coupling between the HC and WC shift modes generate a more prominent springtime (FMA) asymmetry between WNP divergence and southern eastern Pacific convergence (see pink and cyan star symbols). Accordingly, the increased precipitation anomalies in the extratropical regions (e.g., East Asia and South America) are larger for the strong PSYN models than the weak models. These differences in precipitation and the position of the circulation centers are statistically significant at the 99% confidence level during boreal spring but not during boreal winter. We also emphasize that the difference in precipitation is statistically insignificent in the tropics, indicating that the extratropical precipitation response is not a result from a strengthened tropical convective activity (or diabatic heating). We also argue that the difference in model physics may generate a strengthened WC-HC synchronization (i.e., NW-SE skewed asymmetric circulation pattern changes), further intensifying the post-ENSO impact on the extratropical precipitations. We added some discussion on the precipitation impact in the revised text [line 286-296].

[Figure]

**Ref_Fig. 3 Impact of phase synchronized spatial shifts of the HC and WC on global precipitation.** (a) The November-December[0]-January[1] extreme El Niño (i.e., 1982/1983 and 1997/1998) composite anomaly of precipitation (shading) and VP850 (contours) obtained from the (a, c) 8 strong PSYN models in the AMIP simulations and (b) the difference between 8 strong PSYN models and 8 weak PSYN models. (c-d) Same as (a-b), but for February-March-April (FMA[1]). The green dots in (c-d) indicate the area where the difference is statistically significant at the 99% confidence level. The pink and cyan star symbols indicate the positions of maximum and minimum VP850 anomalies for the strong PSYN and weak PSYN models. Here, the strong PSYN and weak PSYN model groups are categorized according to inter-annual correlation coefficients between WC2 and HC1 within 19 AMIP5 and 21 AMIP6 models, respectively (see Supplementary Tables 2 and 3).

**Technical Corrections**

*Q. 1.* Line 29: I would add Kris Karnauskas's 3D Hadley circulation paper to this list: Karnauskas, K.B., Ummenhofer, C.C. On the dynamics of the Hadley circulation and sub- tropical drying. Clim Dyn 42, 2259–2269 (2014). https://doi.org/10.1007/s00382-014- 2129-1

**[Ans]** We added the reference in the list [line 28-29].

*Q. 2. Line 35–36: I would change, ". . .changes during. . ." to ". . .changes, generally during…"*

**[Ans]** We accordingly changed it in the revised test [line 36-37].

*Q. 3. Line 40: this question-in-a-sentence could have its syntax improved.*

**[Ans]** We modified the sentence into "Here we address the question of whether and how WC and HC can also experience concurrent shifts in their geographic positions." in the revised text [line 41-42].

*Q. 4. Line 114: "...there no..." should be "...there is no..."*

**[Ans]** We accordingly changed it in the revised test [line 128].

*Q. 5. Line 125: a bit more intro to the C-mode would be nice.*

**[Ans]** We added more explanation on the C-mode in the revised text [line 137-142].

*Q. 6. Figure 1a, b: The font inside the figures gets small enough as to render it very hard to read in print.*

**[Ans]** We changed the font size largely in Fig. 1a and 1b.

*Q. 7. Figure 1c, d: based on the panel titles, "Interannual CC of...", I thought the plot was showing a running (windowed?) cross-correlation between WC2, HC1, and. . .something else. After flipping back and forth and checking the y-axis, I realized that it was showing the PC time series themselves, thereby illustrating the correlation.*

**[Ans]** We replaced the panel titles into "ERAI (or AMIP MME) PC time series of WC&HC shift mode" in Figs. 1c and d.

*Q. 8. Figure 2 (line 184): be clear that the COLOR of the dots specifies the absolute derivative, while the position of the dots illustrates the phase difference.*

**[Ans]** We modified the explanation more clearly in the caption of Fig. 2.

*Q. 9. Figure 3: the green and blue dots on red are very muddy on my printed pageâA˘T˘and I'm not even colorblind!*

**[Ans]** To clearly classify the dots, we changed the symbol colors and sizes in Fig. 3.

---

## Author Comment (AC2) · 2 Dec 2020

[Research Article # ESD]: "Synchronized spatial shifts of Hadley and Walker circulations" by Kyung-Sook Yun, Axel Timmermann, Malte F. Stuecker

We thank the reviewers for their constructive and helpful comments. We carefully revised the manuscript "Synchronized spatial shifts of Hadley and Walker circulations" and provide a point by point reply to the individual comments below.

**Reply to the comments of Reviewer #2**

**General Remarks:**
*This paper shows that warm pool SST anomalies in the decaying El Niño event gen- erate a meridionally asymmetric Walker circulation response, which couples the zonal and meridional atmospheric overturning circulations. I think their results are overall novel and reasonable, and I would be happy to see this work on Earth System Dynamics after some minor revisions, particularly regarding discussions. Specific comments are as follows.*

**[Ans]** Authors are grateful to the reviewer for very constructive comments which were helpful to improve this work. The manuscript has been largely revised according to the reviewer's comments as listed below.

**Specific Comments:**
*Q. 1. The authors focus on NINO 3 regions to explore the relationship between SST anomalies and Walker circulation. Recently, Central Pacific El Nino events tend to increase and also some papers suggested the increase of CP El Nino in a warm climate. I think that the relationship may be changed if we concentrate on the NINO4 region. The authors need to discuss the sensitivity of the NINO region on the relationship somewhere.*

**[Ans]** The relationship between ENSO and WC&HC shift modes is only manifest in Nino3 but not in Nino4 even during the boreal spring (CC with WC2 ~ 0.09 and CC with HC1 ~ 0.03). This reflects the phase-synchronization is more associated with eastern Pacific-type El Niño events than with central Pacific-type El Niño events. We added the description in the revised text [line 140-142].

*Q. 2. In figure 1, the temporal evolution of normalized PC from WC or HC almost coincides with the NINO3 index from 1979-2000. However, the relationship between the two indexes seems to be weakened after 2000. I would know possible reasons. I think that many reasons may be discussed – ENSO diversity, mult-idecadal variability (IPO or AMO), and even global warming.*

**[Ans]** The weakened relationship may be a consequence of the Pacific WC intensification by Atlantic warming and Pacific cooling (England et al., 2014;McGregor et al., 2014) associated with multi-decadal climate variability (e.g., Atlantic Multi-decadal Oscillation and Inter-decadal Pacific Oscillation), zonal shifts in ENSO's center (Sohn et al., 2013). We added more discussion on the possible reason for the changing WC-HC relationship in the revised text [line 188-191].

***Q. 3.*** *In figure 2, it is well known that the AMIP run tends to overestimate the atmospheric response given SST forcing. How much does the strength or duration length of phase synchronization may be changed in a coupled model?*

**[Ans]** The CMIP5 models show considerable diversity in occurrence probability of SYN event. Compared to nine SYN event from AMIP MME, the number of SYN event from CMIP5 models is ~ 5.7 ± 3.2SD (see Supplementary Fig. S4). We added the explanation in the revised text [line 183-184].

***Q. 4.*** *In Figures 4 and 5, I would see the circulation pattern in the upper troposphere (200 hPa). If the authors think the upper-level circulation change is not relevant to this study, please mention the reason in the main text.*

**[Ans]** We added the 200 hPa circulation pattern as shading in Fig. 4. Basically, the upper-level circulation pattern is consistent with the lower circulation pattern: e.g., anomalous lower-level divergence and upper-level convergence over the WNP. The NW-SE asymmetric circulation feature is slightly clearer in 850 hPa than 200 hPa, likely due to larger impact of SST forcing. We present the 850 hPa circulation pattern only in Fig. 5, to reduce the complexity in plotting.

[Figure]

**Ref_Fig 4 Global pattern of phase synchronized spatial shifts of HC and WC.** Composite anomalies during PSYN months (i.e., the absolute tendency of phase difference is less than 0.3) with February-March-April (FMA[1]) extreme El Niño (FMA[1] Niño3 > 1.5SD), obtained from the observations (left column) and the AMIP MME (right column): (a, b) 850 hPa velocity potential (VP850; contour) and 200 hPa velocity potential (VP200; shading) anomaly; (c-d) precipitation (GPCP for ocean and CRU for land; shading) and 850hPa wind (vector) anomaly. The hatching shows the area where the difference is statistically insignificant at the 99% confidence level.

***Q. 5.*** *In figure 4, the 95 % significance level may be too low to show a strong shift of HC and WC. Why don't you use 99% or other higher criteria?*

**[Ans]** We changed the significance criteria to the 99% confidence level and obtained very similar results. The text was accordingly changed in the caption of Figures.